# The Crystal Structure of the Defense Conferring Rice Protein *Os*JAC1 Reveals a Carbohydrate Binding Site on the Dirigent-like Domain

**DOI:** 10.3390/biom12081126

**Published:** 2022-08-17

**Authors:** Nikolai Huwa, Oliver H. Weiergräber, Alexander V. Fejzagić, Christian Kirsch, Ulrich Schaffrath, Thomas Classen

**Affiliations:** 1Institute for Bioorganic Chemistry, Heinrich Heine University Düsseldorf, 52425 Jülich, Germany; 2Institute of Biological Information Processing 7: Structural Biochemistry and Jülich Centre for Structural Biology, Forschungszentrum Jülich, 52425 Jülich, Germany; 3Institute for Biology III, Department of Plant Physiology, RWTH Aachen University, 52056 Aachen, Germany; 4Institute for Bio- and Geosciences 1: Bioorganic Chemistry, Forschungszentrum Jülich, 52425 Jülich, Germany

**Keywords:** lectin, jacalin-like protein, dirigent-like protein, plant defense

## Abstract

Pesticides are routinely used to prevent severe losses in agriculture. This practice is under debate because of its potential negative environmental impact and selection of resistances in pathogens. Therefore, the development of disease resistant plants is mandatory. It was shown that the rice (*Oryza sativa*) protein *Os*JAC1 enhances resistance against different bacterial and fungal plant pathogens in rice, barley, and wheat. Recently we reported possible carbohydrate interaction partners for both domains of *Os*JAC1 (a jacalin-related lectin (JRL) and a dirigent (DIR) domain), however, a mechanistic understanding of its function is still lacking. Here, we report crystal structures for both individual domains and the complex of galactobiose with the DIR domain, which revealed a new carbohydrate binding motif for DIR proteins. Docking studies of the two domains led to a model of the full-length protein. Our findings offer insights into structure and binding properties of *Os*JAC1 and its possible function in pathogen resistance.

## 1. Introduction

Different abiotic and biotic factors are responsible for global losses in the harvesting of important agricultural crops, such as wheat (29% harvest loss) or rice (40%) [1]. In order to prevent and minimize crop losses, irrigation, fertilizers, and pesticides are used. However, these practices and their potential negative impact on the environment have caused a social debate on reduction or substitution of the classical measures. During the evolution of plants, different strategies to resist the hostility of their environment have been developed. The understanding of such resistance mechanisms against abiotic and biotic stresses and its transfer into modern breeding programs may pave a way to a more environment-friendly agriculture.

A protein involved in resistance against different pathogen groups, which is only present in monocot plants, is referred to as monocot chimeric jacalin (MCJ) [2,3]. As its name suggests, it is a fusion of two proteins, a jacalin-related lectin (JRL) and a dirigent protein (DIR) at the N-terminus. Many representatives of this protein class were found in different monocot plants, such as *As*Crs-1 [4] in creeping bentgrass (*Agrostis stolonifera*), *Pe*D-J [5] in Moso bamboo (*Phyllostachys edulis*), *Ta*Ver2 [6] and *Ta*HFR1 [7] in wheat (*Tri**ticum aestivum*), *Sb*SL [8] in sorghum (*Sorghum bicolor*), *Zm*BGAF [9] in maize (*Zea mays*), *Sh*DJ [10] in sugarcane (*Saccharum hybrid cultivar*), and *Os*JAC1 [11] in rice (*Oryza sativa*). Recently, Ma and Han (2021) identified 46 MCJ genes from the wheat genome and divided the class into three subfamilies [12]. Consistent with the literature, their findings showed the functional diversity of MCJs, which are able to respond to different biotic and abiotic stress factors. Accordingly, one representative of subclass 3 (MCJ3) was able to increase resistance against drought [12]. The homologue *Sh*DJ from sugarcane also showed drought resistance properties along with the ability to improve biomass production and to increase saccharification [10]. Another feature previously observed for some MCJs is the protein–protein interaction with β-glucosidase [9]. However, the physiological function behind this MCJ capacity is still unknown. More well-described properties of subfamily MCJ1 include the broad-spectrum pathogen resistance conferred by *Os*JAC1 [13] or *Ta*JA1 [14]. Among the two MCJ domains, the JRL domain is better characterized, particularly regarding its carbohydrate binding properties. Previously identified interactions displayed specificity toward mannose [7,11,12,15], galactose [9,12], *N*-acetylgalactosamine [8], or *N*-acetylglucosamine [6].

In contrast, there is little physiological information available about the properties of the DIR domains in the full-length MCJs. So far, it was reported that after the deletion of the DIR-domain the agglutination activity of *Zm*BGAF was abolished [9], the sugar binding pattern was altered [16], and the pathogen resistance was lost [13]. However, it is still not known how the DIR domains in MCJs add to pathogen resistance. Generally, DIR proteins are involved in provision of different secondary metabolites in plants, e.g., phenolic intermediates for the lignan synthesis [17,18], aromatic terpenoids for the defense response against herbivorous insects and pathogens [19], as well as pterocarpans acting as phytoalexins [20,21]. Furthermore, DIR proteins are involved in the abiotic stress response [22,23] and take part in the formation of Casparian strips of root tissue [24]. However, the biochemical function of the majority of DIR proteins is still unknown [20]. Nevertheless, it is assumed that DIR domains in MCJs share biochemical activities with their non-chimeric homologs.

Recently, we characterized both domains of *Os*JAC1 biochemically, revealing new binding properties in either case [25]. We demonstrated that the DIR domain is able to recognize and selectively bind carbohydrates, similar to the JRL domain [25]. This was a novel finding because known DIR proteins are involved in oxidative processes and have auxiliary enzymatic functions [26] but are devoid of sugar binding activities. To further study the properties of *Os*JAC1, we have now determined the crystal structures of its two domains. These structures enabled in silico docking studies giving insights into the architecture of the full-length protein. Furthermore, we were able to obtain a complex structure featuring the DIR domain and a putative carbohydrate binding partner (galactobiose), which revealed the binding site. Finally, the interaction with cell surface carbohydrates was investigated through hemagglutination assays for the full-length protein and its two domains.

## 2. Methods and Materials

### 2.1. Materials

*Os*JAC1 and its two individual domains were prepared as previously described in Huwa et al. (2021) [25]. The initial crystallization screening conditions of the two domains that led to successful crystallizations, are listed in Table 1. All solutions were prepared or diluted with ddH_2_O. Laminaribiose and 1,4-β-d-galactobiose were purchased from Megazyme (Bray, Ireland). Dithiothreitol (DTT) was purchased from Fisher Chemical (Schwerte, Germany). Rabbit blood in Alsever’s solution was purchased from Fiebig-Nährstofftechnik GbR (Idstein-Niederauroff, Germany). Screening sets for protein crystallization were purchased from Merck (Darmstadt, Germany) and Qiagen (Hilden, Germany).

### 2.2. Crystallization, Soaking Experiment and Data Collection

Prior to crystallization experiments, the DIR domain His-tag was removed by incubation with thrombin (2 U; substrate 6.25 mg/mL, Cytiva, Marlborough, MA, USA) at room temperature for 48 h. Quantitative cleavage was verified by SDS-PAGE and no further purification steps were conducted. Crystallization screening for the two individual *Os*JAC1 domains was performed at 20 °C on a Freedom Evo robotic system (Tecan, Männedorf, Switzerland) in sitting-drop geometry using a 2:1 ratio of protein stock and reservoir solution. Screening conditions which gave rise to the crystals reported in this work are listed in Table 1. Crystals of the DIR domain with bound galactobiose were obtained by soaking preformed crystals of the unliganded DIR domain (cf2) in the respective reservoir buffer containing 1 mM galactobiose. Crystals were harvested after 1 h of soaking time. Single-wavelength diffraction datasets of *Os*JAC1 DIR domain samples were recorded at 100 K on beamline ID30A-3 (*Os*JAC1-DIR) and beamline ID23-1 (*Os*JAC1-DIR with galactobiose) of the European Synchrotron Radiation Facility (ESRF; Grenoble, France). The *Os*JAC1-JRL datasets were acquired at 100 K on ESRF beamlines ID23-1 and ID29. For data collection statistics, refer to Table 1. The recorded data sets were processed using XDS and XSCALE [27].

### 2.3. Structure Determination and Refinement

All X-ray structures presented in this work were determined by molecular replacement. For the two crystal forms of *Os*JAC1-DIR, independent solutions based on the structure of PTS1 from *Pisum sativum* (PDB-ID 6OOD) were found by applying MoRDa [28]. An initial model for *Os*JAC1-JRL-cf1 was obtained with MOLREP [29] using the structure of banana lectin (PDB-ID 4PIF); following initial refinement this model served as a molecular replacement template for JRL-cf2. Models were improved iteratively by alternating reciprocal space refinement in phenix.refine [30] with interactive rebuilding in Coot [31]. Final atomic coordinates and structure factor amplitudes have been deposited in the Protein Data Bank and the access codes are listed in Table 1, along with the refinement statistics. All structural figures were generated with UCSF Chimera [32], unless indicated otherwise.

### 2.4. Ligand-Protein and Protein-Protein Docking

Ligand–protein docking was conducted with AutoDock Vina [33]. Prior to docking, hydrogen and Gasteiger charges were added to the respective 3D domain structures. Chem3D was used to build and energy-minimize the ligand structures (sugars) and to create the mol2 format of the ligand descriptions. Open Babel 2.3.1 was used to reformate the ligand file toward pdbqt. The two binding sites of the JRL domain (pdb 7YWG/chain A; site I and II) were implemented separately for docking with position 13.7, 46.4, and 24.0 Å (*x,y,z*) for putative binding site I and position 38.5, 46.4, and 24.0 Å (*x,y,z*) for putative binding site II; box dimensions used were 24.8 × 28.1 × 21.98 = 15,310 Å³.

Domain interaction was analyzed in silico by docking both domains through the web-based docking program HADDOCK V2.4 [34]. Prior to the domain docking, possible interaction residues for each domain were predicted with the web tool CPORT [35]; refer to Appendix A for a complete list. All default docking parameters from HADDOCK V2.4 were applied. Schematic representations of ligand–protein or protein–protein interactions were prepared using LigPlot^+^ [36].

### 2.5. Hemagglutination Assay

The hemagglutination assay was conducted based on the protocol of [37] with slight modifications. In the initial step, the erythrocytes were washed at 4 °C with cooled PBS (phosphate-buffered saline; 150 mM NaCl, 20 mM Na-phosphate, pH 7.2). The suspension was centrifuged at 500× *g* for 5 min at 4 °C, followed by removal of the supernatant. The washing step was repeated three times and/or until the supernatant was clear. For the determination of the hemagglutination activity, the proteins (*Os*JAC1 and its domains) were serially diluted in a 96-well microtiter plate (*Greiner Bio One*™, V-shape bottom, *Greiner Bio-One* GmbH, Frickenhausen, Germany). The final protein concentration of the first row for each protein was 0.5 mg/mL. The assay was conducted with a 1% (*v*/*v*) rabbit erythrocyte suspension. The results of the assay were documented after 1 h incubation at room temperature by visual determination of the hemagglutination. The following controls were included: buffer without any protein as negative control and the *Os*JAC1-JRL domain as positive control. The *Os*JAC1-JRL domain has previously been shown to agglutinate rabbit erythrocytes [11].

### 2.6. Growth Inhibition Assay

The assay was conducted in triplicate on a 96 × 300 µL microtiter plate (polystyrene, transparent, flat bottom; *Greiner Bio-One* GmbH, Frickenhausen, Germany). LB media containing 0.1 mM ampicillin was inoculated towards an OD_600_ of 0.1 with an overnight culture of *Escherichia coli* BL21(DE3). *Os*JAC1 was serially diluted from 3.5 mg/mL to 27 µg/mL. The total assay volume was 150 µL (135 µL LB media + 15 µL protein solution). The plate was sealed with an air-permeable foil (*AeraSeal*^TM^, Excel Scientific, Vicorville, CA, USA) and incubated on a plate shaker for 4 h at 30 °C and 800 rpm. After the incubation, the OD_600_ was determined by an Infinite M1000 Pro plate reader (Tecan, Männedorf, Switzerland). Following subtraction of control absorbance, OD_600_ values of the culture were plotted against the substrate concentration.

## 3. Results

### 3.1. Overall Structure of the OsJAC1 Domains

Efforts to crystallize the full-length MCJ protein *Os*JAC1 did not yield any hits during the screening of over 2000 different conditions. However, the individual domains of *Os*JAC1, JRL, and DIR, were each crystallized in two different crystal forms. The X-ray structures of both domains were determined by molecular replacement. In addition to the apoprotein structure of the DIR domain, we were able to obtain a complex structure containing galactobiose as a ligand, which was previously shown to bind to the DIR domain using differential scanning fluorimetry and circular dichroism spectroscopy [25]. The X-ray crystallographic statistics are summarized in Table 2.

The two domains of *Os*JAC1 show characteristic features of their corresponding protein families (refer to Appendix A for a schematic of the full-length protein along with its amino acid sequence). The JRL domain has a β-prism-I fold composed of three four-stranded antiparallel β-sheets. Two of these sheets are canonical Greek key motifs, whereas in the third case the very N- and C-terminal segments of the protein join to form the outer and inner pair of strands, respectively. In both space groups described in this work, the JRL protein crystallizes with two chains per asymmetric unit, which interact in the same manner (Figure 1A; PDB: 7YWG). The two dimeric assemblies of *Os*JAC1-JRL superpose with a root-mean-square (RMS) distance of 0.87 Å for Cα atoms and 1.24 Å for all non-hydrogen atoms (using residue range 160–306 in all subunits). The dimer itself features C_2_ symmetry with constituent chains being more similar in crystal form (cf) 1 of the *Os*JAC1-JRL domain (Cα RMS distance 0.58 Å) than in JRL-cf2 (Cα RMS distance 1.13 Å). As indicated by PISA analysis [38], the extensive interface involving both hydrophobic and polar interactions buries more than 800 Å^2^ of solvent-accessible surface area on either protomer and is classified as stable in solution. The structure of the JRL domain of *Os*JAC1 is highly conserved and shares structural similarities with classical mannose-selective JRLs (mJRLs), such as artocarpin from *Moraceae* [39] or frutalin from *Artocarpus incisa* (PDB entry 4WOG), with a sequence similarity of approximately 30%. In contrast to all other mJRLs from plants, the two chains of the non-crystallographic dimer of *Os*JAC1-JRL are linked via a symmetric disulfide bond formed by a cysteine residue close to the N-terminus (Cys161). So far, only for the JRL homolog in the pearl shell (*Pteria penguin*) such an inter-subunit disulfide bond has been reported, but in this case bond formation was mediated by cysteines at the C-termini [40]. The possibility of disulfide bond formation for the JRL domain was already indicated in our previous work based on a shift in the thermal melting point upon oxidation [25]. Overall, the intimate interaction between subunits suggests a physiologically relevant mode of dimerization; accordingly, the disulfide bond may not just represent an artefact of crystallization under non-reducing conditions but may actually occur in the full-length protein (see Discussion section).

The isolated DIR domain of *Os*JAC1 presents as a trimer in the asymmetric unit of both crystal forms (see Figure 1B), featuring C_3_-symmetry with the same mode of interaction among constituent chains. In fact, the two trimers are slightly more similar than the JRL domain dimers, superposing with RMS distances of 0.64 and 0.95 Å for Cα atoms and all atoms, respectively. It should be noted that the trimeric state has been consistently observed for DIR and DIR-like proteins after crystallization [20,30,31]. All *Os*JAC1 DIR protomers show the canonical eight-stranded antiparallel β-barrel fold (Figure 1C) known from structurally characterized homologs [20]. The length of the traced chain differs significantly between the molecules of the DIR trimers, particularly in the N-terminal regions. These show signs of enhanced dynamics, exhibiting different degrees of disorder in the crystal structures. Consequently, the average pairwise Cα RMS distance between chains, amounting to 0.79 and 0.90 Å for DIR-cf1 and -cf2, respectively, reduces to 0.45 and 0.39 Å if residues preceding Leu26 or following Cys153 are omitted. In either model, chain B has the highest amino acid sequence coverage. Interestingly, residues Lys6–Thr10 were found hydrogen-bonded to Gln20–Thr24 of an adjacent chain (chain A in cf1) in the same trimer, adding an extra antiparallel strand to the neighboring β-barrel. The distal N-terminal segments of the other chains could not be traced and thus do not reveal the same type of strand swapping. Since the chains of a trimer are a priori equivalent, this discrepancy is likely to result from differences in the lattice environment, rather than representing an intrinsic asymmetry. Similar to the JRL domain dimer, the DIR domain trimer is categorized as a stable assembly by PISA [38]; the solvent-accessible surface buried at the binary interfaces amounts to approximately 900 Å^2^ per subunit, increasing to above 1400 Å^2^ if the swapped β-strand is considered. Structural alignment of the *Os*JAC1-DIR domain with homologs, such as *At*DIR6 (PDB: 5LAL) and *Ps*PTS1 (PDB: 5OOD), revealed RMS distances of approximately 1 Å for the core β-barrel structure. Increased deviations are observed for the loops as well as the N- and C-terminal regions of the proteins. In particular, the loops lining the substrate binding sites differ in their lengths and conformations (see Appendix A). While these loops constitute a wide-open binding area in the *Os*JAC1-DIR domain, they form a binding cavity in *At*DIR6 [41] and a deep putative active-site pocket in *Ps*PTS1 [20].

### 3.2. Carbohydrate Binding Site of the OsJAC1-DIR Domain

Known dirigent proteins are associated with the auxiliary conversion of phenolic compounds, such as lignans or terpenoid phenols [17,19]. The functionalities of DIR domains in MCJs have to the best of our knowledge never been reported. Here, we show the binding site of galactobiose within the DIR domain of *Os*JAC1 (see Figure 2).

Complex crystals were obtained by incubating specimens of *Os*JAC1-DIR crystal form 2 with the β(1⟶4) variant of galactobiose, which we had previously identified as a high-affinity *Os*JAC1 ligand. The soaking procedure resulted in an overall increase of disorder, as evidenced by elevated B-factors and a reduction of useable resolution. Nevertheless, the disaccharide could be clearly identified by newly appearing difference density in all three chains of the DIR domain trimer (see Figure 2A for a-posteriori ligand verification). The binding site is located on the pole of the barrel opposite the N- and C-terminus, and consists of a widely accessible pocket, which is flanked by the differently sized loops L1, L2, L4, L6, and L8 (see Appendix A). In this groove, the non-reducing unit of the galactobiose is precisely fitting and is able to interact through hydrogen bonding (prominently with His30 and Gln39, more weakly—as judged by non-hydrogen atom distances >3.2 Å in two out of three instances—with Glu111), and through van der Waals contacts (involving Pro32, His79, Trp85, and Val110; see Figure 2B and Appendix A), on average burying 150 Å^2^ of solvent-accessible protein surface area. The reducing unit exhibits less contact with the protein, forming hydrogen bonds mainly with Glu111 and Asn143, which corresponds to approximately 90 Å^2^ of buried protein surface. The results of refinement indicate that ligand occupancy is close to stoichiometric for all three binding sites. Notably though, comparison with the structure of the original *Os*JAC1-DIR crystal form 2 did not reveal major conformational changes related to binding of the disaccharide.

Figure 3 shows a sequence alignment of the DIR domain of *Os*JAC1 with different known MCJ and DIR proteins. The yellow bars correspond to the loops and the grey arrows to the β-strands of the *Os*JAC1-DIR domain. The investigation of potential ligand binding motifs based on those described for *Ps*PTS1 [20] and *Os*JAC1 (this study) revealed two different conserved patterns (indicated by red and black frames in Figure 3). Specifically, BLAST searches with different algorithms (BLASTP, PSI-BLAST, and DELTA-BLAST [45,46]) showed that the carbohydrate-binding motif of the *Os*JAC1-DIR domain is mostly found in MCJ proteins and is highly conserved in this family, especially for the subgroup MCJ1 defined by Ma and Han (2021) [12]; *At*DIR6, on the other hand, contains a signature similar to that in *Ps*PTS1. These observations suggest that the DIR domains of the MCJ1 subgroup not only exhibit the same tertiary structure as the *Os*JAC1-DIR domain but are likely to also feature similar carbohydrate binding properties. *Ta*MCJ2 and *Ta*MCJ3 exhibit a slightly different sequence pattern at the binding area of the *Os*JAC1-DIR domain, which indicates a more distant evolutionary relation and possibly a different function.

### 3.3. Carbohydrate Binding Site of the OsJAC1-JRL Domain

Classical JRLs are categorized according to their binding selectivity to either mannose (mJRL) or galactose (gJRL) [48], and their general binding motif is well described [49]. The JRL domain of *Os*JAC1 is selective for mannose [11]. A higher melting point increase upon binding of the disaccharides laminaribiose and 2α-mannobiose compared to the corresponding monosaccharides was determined via differential scanning fluorimetry [25].

In order to investigate the binding site of the *Os*JAC1-JRL domain, docking experiments with laminaribiose were performed. The described binding sites of Banlec [50] were used as target areas. Figure 4 shows selected docking results with laminaribiose and the respective interactions involved.

We were able to identify two small, separated carbohydrate-binding pockets (Figure 4A), which share some similarities with each other. Each pocket is formed through two loop regions emerging from one antiparallel β-sheet Greek key motif. In fact, these two carbohydrate-binding sites and the interactions involved are quite similar to what has been found for Banlec [50]. A structural alignment yields an RMS distance between the Cα atoms of the *Os*JAC1-JRL domain and Banlec (PDB: 2BN0) of 2.1 Å. According to Meagher et al. (2005) [50], the two carbohydrate-binding sites of Banlec (sites I and II) each consist of one Gly-Gly loop and a Gly-X_3_-Asp motif loop. The docking of laminaribiose toward site II of the *Os*JAC1-JRL domain involves an interaction with both mentioned loops (Gly222–Gly223; Gly135–Asp139); additionally, Asp224 and Tyr295 might interact with the ligand and stabilize it further (see Figure 4C). The docking of laminaribiose to the first potential binding site of *Os*JAC1-JRL domain (site I) shows a different interaction pattern (see Figure 4B). Site I of the *Os*JAC1-JRL domain has a Gly-X_3_-Asn motif loop (residues 293–297) mediating similar interactions as observed for site II. However, in the other loop of site I, one glycine of the Gly-Gly motif is substituted with a glutamic acid (Glu174), which might alter the ligand interaction pattern and orientation compared to site II.

Figure 5 shows a sequence alignment of the *Os*JAC1-JRL domain with different MCJ and mJRL proteins. The bars correspond to the loops and the arrows to the β-strands of the *Os*JAC1-JRL domain. It should be noted that the automatic assignment of β9 is imperfect in *Os*JAC1-JRL, although the angles φ and ψ are in the Ramachandran area of β-strands. The residues of sites I and II described above are framed in black and yellow, respectively. The highest sequence similarity in regard to the proposed ligand binding motif of *Os*JAC1-JRL is found for the proteins *Ta*MCJ1, *Sh*DJ, and *Acm*JRL (homolog of Banlec). These three proteins should thus exhibit two carbohydrate binding sites as already shown for *Acm*JRL [51]. On the other hand, *Ta*MCJ3 and SalT might be monovalent using site II only, which was shown for SalT [52]. *Ta*MCJ2 does not share the above-described binding motif and might belong to another subclade of JRL proteins.

### 3.4. Domain-Domain Interaction Analysis

Despite extensive trials, we did not observe crystals of the full-length *Os*JAC1 protein. To identify if or how both domains are able to interact with each other, we therefore conducted an in silico analysis by docking both domains through the web-based docking program HADDOCK [34]. Prior to the domain docking, possible interaction residues for each domain were predicted with the web tool CPORT [35]. Predicted active residues for the JRL domain included residues of strands β1 and β10 (see Figure 1A). However, those two β-strands constitute the primary dimerization interface of JRL protomers, which are connected by the inter-subunit disulfide bond (Cys161-Cys161) in our crystal structures. Formation of an inter-subunit disulfide bond is also supported by our previous observations with the full-length protein [25], and we hypothesize that Cys161 is instrumental to the physiological role of *Os*JAC1. Hence, we excluded these antiparallel β-strands of the JRL domain from docking with the DIR domain. All other predicted active residues were supplied to the HADDOCK algorithm as potential attractors and are listed in Appendix A. Figure 6 shows the most favorable pose, based on chain A of crystal form 1 of both *Os*JAC1 domains. This model, which essentially represents the full-length *Os*JAC1 protein, was subsequently subjected to an additional round of docking, in order to determine potential modes of self-association (Figure 7).

The HADDOCK model of the two-domain *Os*JAC1 protein (Figure 6) features an extensive interface, burying more than 2100 Å^2^ of solvent-accessible surface. Interaction is mainly mediated by strands β2 and β5 of the JRL domain and loops L2 and L7 as well as strands β2–β6 of the DIR domain, and predominantly involves nonpolar amino acids (see Appendix A for a schematic representation). Importantly, the C-terminus of the DIR domain is placed in close proximity to the N-terminus of the JRL domain, consistent with the (unmodeled) four-residue connecting segment. As a result, the ligand binding sites of the two domains (yellow for JRL and blue for DIR), which are located opposite their respective termini, are facing the same direction.

The different oligomeric states of the individual domains in the crystallographic asymmetric units (Figure 1A,B) were considered as templates for additional docking experiments, assessing potential self-association of the full-length *Os*JAC1 protein (see Figure 7A,B, for scoring statistics refer to Appendix A). The two-domain model described above was found to be perfectly compatible with the JRL–JRL interaction mode observed in the respective crystal structures. For instance, superposition of the JRL-cf1 homo-dimer and the corresponding HADDOCK model of dimeric full-length *Os*JAC1 (see Figure 2A) yielded an RMS distance of 0.56 Å for all Cα atoms (using residue range 160–306 in all subunits; Appendix A). Besides the JRL interface, oligomerization of the full-length protein might be mediated by self-interaction of the DIR domain. The individual DIR domain appeared as a trimer in the crystallographic asymmetric unit (Figure 1B) burying many hydrophobic residues on the protomer surfaces (Appendix A). Since this trimer is incompatible with the domain arrangement proposed for the full-length protein, we chose to investigate potential dimer states that retain as much of the DIR trimer interface as possible. Figure 7B illustrates a dimeric arrangement of the full-length *Os*JAC1 protein that facilitates hydrophobic interactions of the DIR domains in the center of the complex but also involves contact between JRL and DIR domains of different subunits in the periphery (Appendix A). Importantly, the two modes of dimerization illustrated in Figure 7 are not mutually exclusive; provided that both are indeed feasible, this would enable even higher oligomeric states. Thus, the true oligomeric state remains elusive and needs further experiments for clarification. Thus, the true oligomeric state remains elusive and its clarification needs further experimental evidence, e.g. from small-angle X-ray scattering or analytical ultracentrifugation measurements using the full-length protein.

### 3.5. Hemagglutination Assay

Lectins are typically able to interact with the carbohydrates on the surface of erythrocytes, which leads to agglutination of red blood cells. We have used a hemagglutination assay with rabbit erythrocytes to quantitate the lectin activities of full-length *Os*JAC1, as well as its isolated domains. Table 3 summarizes the results; pictures of the plates for the assay are provided in the Appendix A.

The full-length *Os*JAC1 protein and the DIR domain showed a similar and a strong response toward the tested rabbit erythrocytes. The lowest protein concentration required for hemagglutination was approximately 1 µg/mL for both proteins, with no observed difference regarding the oxidative state of the assay. The separated JRL domain of *Os*JAC1 showed a weaker response toward the rabbit erythrocytes. Under oxidizing conditions, minimum concentrations between 1.95 and 3.9 µg/mL were determined, which corresponds to previously reported values (1.95 µg/mL) [11]. Intriguingly, the hemagglutination activity of the separated JRL domain decreased up to 8-fold under reducing assay conditions (4 mM DTT). These observations suggest that, in contrast to full-length *Os*JAC1 and the DIR domain, the JRL domain relies on dimerization via the inter-subunit disulfide bond to achieve optimal hemagglutination activity.

## 4. Discussion

Monocot chimeric jacalins are reported to be involved in different plant stress responses, such as drought [10], or pathogen resistance, such as against powdery mildew [13], which raised interest for a potential use in agriculture. However, despite increased research efforts, our knowledge about their mechanisms in the plant cell is still scarce. In our previous work, we established the requirements for an in-depth biochemical analysis of the MCJ *Os*JAC1 and reported new properties, including the binding of galactose containing sugars to the DIR domain [25]. The galactose-containing disaccharides have been picked as model substrates in an initial screening [25], and they have shown superior binding effects for the dirigent domain. Similar results had the diglucoside laminaribiose and mannobiose. It should be noted here that these disaccharides are presumably not the native ligands of the proteins because there is no evidence for an infection-related function of these disaccharides. Rather, we assume that they are surrogates for larger glycan structures that are part of glycosylated proteins or the cell wall. Components either of the pathogen, which penetrates the plant cell wall using appressoria, or plant components, which are made accessible due to the penetration process, come into play. Endeavors to identify the native ligand by pull-down/fishing-experiments were, as of yet, not successful. In this report, we expand our understanding of *Os*JAC1 by providing the first crystal structures of its two domains. The domains showed the characteristic architectures of their respective protein families, i.e., the β-prism-I fold for JRLs [53] or the β-barrel fold for DIR proteins [54]. Based on the docking results and the hemagglutination assay, the *Os*JAC1-JRL domain is likely to exhibit two closely apposed carbohydrate-binding sites. The increased hemagglutination activity of the *Os*JAC1-JRL domain under oxidizing conditions points to a crucial role of the intermolecular disulfide bond that is consistently observed in both crystal forms. In the absence of a covalent linkage the multivalency required for hemagglutination may be achieved either via the existence of duplicate binding sites, the short distance of which may limit efficiency, or by transient dimerization. A disulfide-bridged JRL dimer, on the other hand, would constitute a stable tetravalent carbohydrate binder with favorable distance between interaction sites, which would be expected to promote networking between carbohydrates of the erythrocytes, as illustrated in Figure 8.

The *Os*JAC1-DIR domain revealed an eight-stranded antiparallel β-barrel fold with a wide-open putative binding area, which is surrounded by five loops (see Figure 2). Interestingly, these loops are the sites of highest structural differences between DIR homologs (Appendix A). In the case of canonical DIR domains, the diversity of the putative active/binding sites might be reflected in different produced phenolic compounds, such as lignans [18], neolignans [55], pterocarpans [20] or terpenoid phenols [19]. In contrast, the *Os*JAC1-DIR domain showed a high binding preference for disaccharides containing galactose as the non-reducing unit. The carbohydrate-binding motif of *Os*JAC1-DIR was identified predominately in DIR domains of MCJs and has so far not been predicted in stand-alone DIR proteins. During evolution, the fusion of two genes to a new entity coding for a chimeric protein may have given rise to neofunctionalization and the original genes might have gotten lost [56]. The biological roles of DIR domains in *Os*JAC1 and related MCJs could therefore differ from those of other family members and may need to be reconsidered.

Figure 9 illustrates potential contributions of *Os*JAC1 to pathogen defense, which include a direct inhibitory effect, recruitment of other defense-related proteins, and auxiliary enzyme activity. Previous studies showed that in transgenic barley, *Os*JAC1 accumulates at the infection site during fungal attack, which was also the case for the single JRL but not for the DIR domain [13], indicating that the JRL domain is essential for targeting *Os*JAC1 to its presumed site of action. Whether the JRL domain is able to recognize glycoconjugates related to the membrane or glycoproteins from the plant or the fungus, is still unknown. However, our previous observations exclude glycoconjugates containing cellobiose or *N*,*N*′-diacetyl-chitobiose, whereas glycoconjugates containing mannobiose (α1-2) or laminaribiose (β1-3) emerged as favorable target structures [25]. The potential inhibitory effect of the DIR domain in the defense mechanism might include the direct binding to pathogen related targets (via carbohydrate moieties) and their subsequent inactivation through the formation of agglutinated particles. Moreover, it was shown that *Os*JAC1 is involved in the abiotic stress response to DNA damage from gamma radiation [57] and hence should be able to also interact with proteins of the plant cell. *N*- and *O*-glycans of plant proteins contain, among other sugars, also galactose [58], which could be captured by the DIR domain, suggesting a mechanism for recruitment of other defense-related proteins. Nevertheless, it cannot be ruled out that the DIR domain is involved in the production of secondary metabolites for pathogen defense; in this scenario, galactobiose may be a surrogate replicating certain properties of the physiological substrate. Homogenized leaf material of *Os*JAC1-overexpressing plants showed a growth inhibiting effect on *Xanthomonas oryzae* pv. *oryzae* and *Escherichia coli* [13], whereas heterologously produced and isolated *Os*JAC1 did not result in any growth inhibition of *Escherichia coli* (see Appendix A). These results suggest that a necessary factor is missing in recombinant *Os*JAC1 to exert its antibacterial properties, and a yet-to-be-identified DIR domain substrate represents an obvious candidate. Moreover, either the DIR domain or the substrate may need to be activated. As an example of the first case, the DIR domain might only have its auxiliary enzyme activity upon binding of the JRL domain to its target. Recall that, according to our docking model, loop L2 of the DIR domain is positioned close to the carbohydrate binding sites of the JRL domain (see Figure 6) and might thereby sense their engagement. A requirement for substrate activation, on the other hand, is a defining property of DIR proteins, as described for *At*DIR6 [41], *Ph*DIR [59], or *Gh*DIR4 [19], which guide the coupling of radical activated phenolic substrates. During the pathogen attack reactive oxygen species are formed in the plant cell [60]; these change the redox state of the cell and might thus play a role not only in formation of the disulfide-linkage between *Os*JAC1 JRL domains but also for activation of a putative DIR domain substrate.

We were looking into possible oligomerization of the full-length protein. This analysis resulted in two plausible dimer models (Figure 7A,B). We note that the combination of both dimerization modes would enable formation of oligomers extending into long chains (fibrils); interestingly, a tetramer with an approximate molecular mass of 140 kDa (Appendix A) would match the results previously obtained by dynamic light scattering under reducing conditions [25]. The coexistence of different oligomeric states also offers a plausible explanation for our difficulties in crystallizing the full-length protein. Overall, the three-dimensional structures of *Os*JAC1 presented in this work, obtained by X-ray crystallography and in silico docking, provide us with a solid basis for understanding the protein’s architecture, which is crucial for follow-up studies to elucidate the relevant function of *Os*JAC1 and other chimeric jacalins in the pathogen resistance response.

## Figures and Tables

**Figure 1 biomolecules-12-01126-f001:**
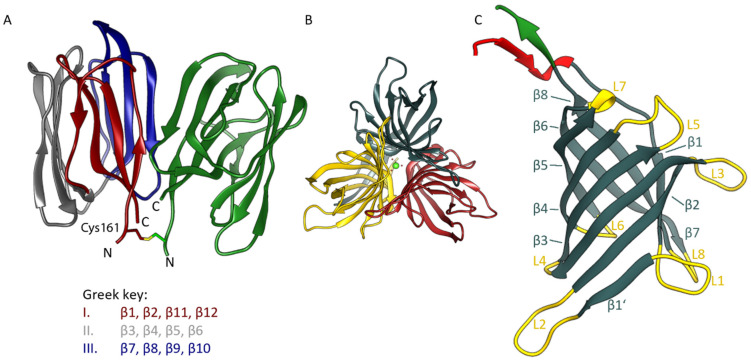
Structure of *Os*JAC1 domain proteins in ribbon representation. (**A**) JRL dimer as found in the asymmetric unit of both crystal forms, with one chain colored in grey, red, and blue, indicating the three four-stranded antiparallel β-sheets (Greek key motifs I–III), the other monomer is coloured green. The figure is based on coordinates from crystal form 1; N- and C-termini (at residues Gln160 and Pro305, respectively, of full-length *Os*JAC1) are labelled. The homodimer features a symmetric inter-subunit disulfide bond at Cys161. (**B**) DIR trimer constituting the asymmetric unit of both crystal forms. The figure was prepared using coordinates from crystal form 1, including a calcium ion (green) in the center. (**C**) β-barrel architecture of the DIR monomer (crystal form 1, chain B). Loops (labelled L1 through L8) are highlighted in yellow and β-strands (labelled β1 through β8) shaded in metallic. The adjacent N- and C-terminal segments are colored red and green, respectively.

**Figure 2 biomolecules-12-01126-f002:**
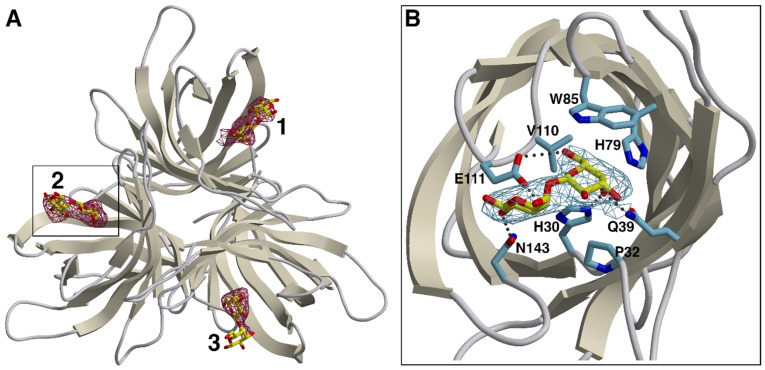
Binding of galactobiose to the *Os*JAC1 DIR domain. (**A**) Simulated-annealing omit map verifying the presence of the ligand. Following removal of the three galactobiose instances from the coordinate file, the model was subjected to an additional refinement run including simulated annealing (starting temperature 3000 K). The resulting difference density (dark red, *mF_o_-DF_c_* synthesis) is contoured at 2.5 RMSD in the proposed carbohydrate binding sites (labeled 1, 2, and 3 for protein chains A, B, and C, respectively). The DIR trimer (ribbons) and the galactobiose molecules (stick models, carbon atoms in yellow) represent the deposited coordinates. (**B**) Close-up of site 2, viewed along the central axis of the respective DIR domain β-barrel (other subunits omitted for clarity). Protein side chains interacting with the galactobiose ligand via hydrogen bonds (dotted lines) or van der Waals contacts are indicated. Here, the electron density (stick representation, carbon in light blue) corresponds to the *2mF_o_-DF_c_* map (blue) obtained for the complete model. The figure was created using POVScript+ [42] and Raster3D [43], using secondary structure assignments generated by DSSP [44].

**Figure 3 biomolecules-12-01126-f003:**
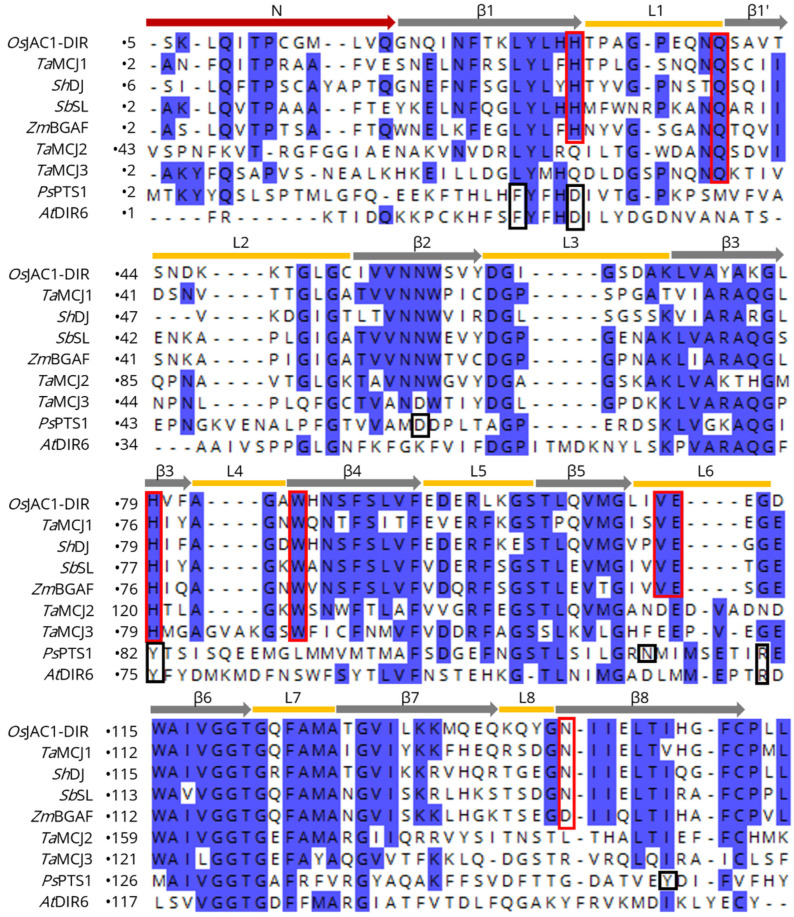
Sequence alignment (CLUSTAL OMEGA multiple sequence alignment [47]) of the *Os*JAC1-DIR domain with different MCJ and DIR proteins. Yellow bars correspond to the loops, grey arrows to the β-strands, red arrow to the N-terminal regions of the *Os*JAC1-DIR domain; the carbohydrate binding residues described in this work are highlighted by red frames. Black frames indicate residues of the active site in *Ps*PTS1 [20]. Conserved regions (>50% identity) are shaded in blue. The sequences correspond to proteins from *Triticum aestivum* (*Ta*MCJ1, -2, -3), *Saccharum hybrid* (*Sh*DJ), *Sorghum bicolor* (*Sb*SL), *Zea mays* (*Zm*BGAF), *Picea sitchensis* (*Ps*PTS1), and *Arabidopsis thaliana* (*At*DIR6).

**Figure 4 biomolecules-12-01126-f004:**
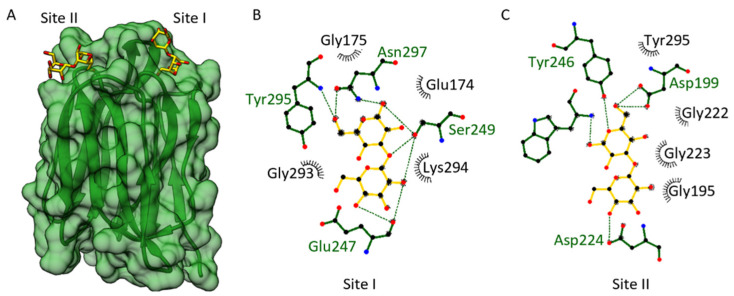
Docking of laminaribiose to the JRL domain of *Os*JAC1. (**A**) Transparent surface view of *Os*JAC1-JRL (cf1 chain A, green ribbon) with the two proposed binding sites (site I and site II) for laminaribiose (stick representation, carbon colored yellow). (**B**,**C**) Ligand interaction schemes (generated using LigPlot^+^ [36]), indicating the residues involved in site I and site II, respectively. Hydrogen bonds are shown in green and hydrophobic interactions in black. The ligand laminaribiose is represended in yellow bonds, protein residues are shown in green.

**Figure 5 biomolecules-12-01126-f005:**
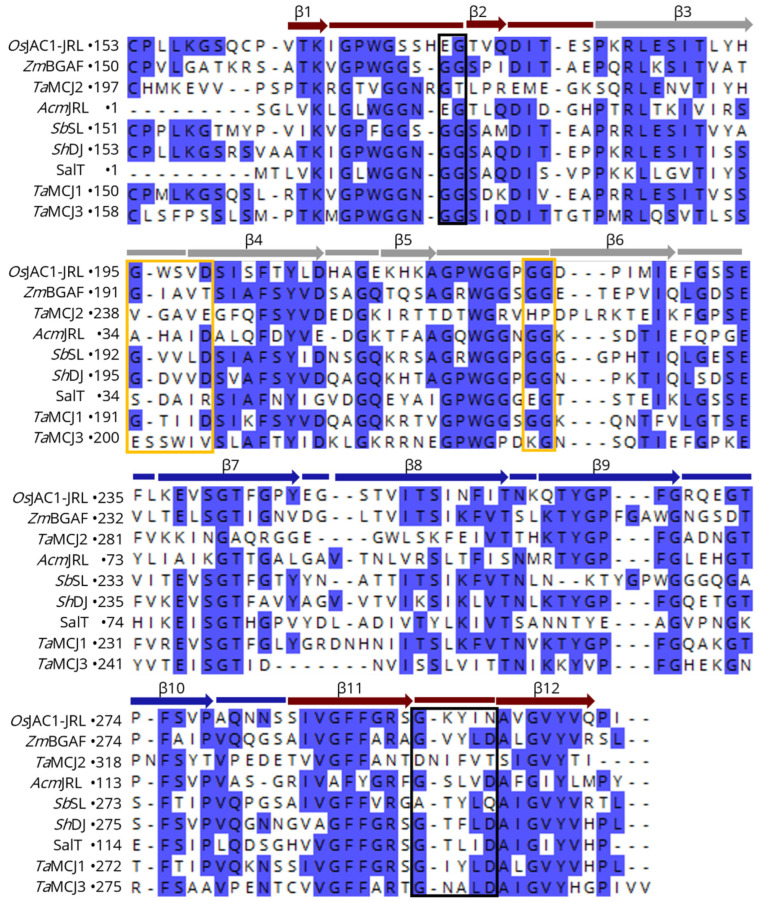
Sequence alignment (CLUSTAL OMEGA multiple sequence alignment [47]) of the *Os*JAC1-JRL domain with different MCJ and mJRL proteins. Bars correspond to the loops and arrows to the β-strands of the *Os*JAC1-JRL domain (coloring indicates the three four-stranded antiparallel β sheets (Greek key motifs I–III) as in Figure 1A). Black and yellow frames denote residues of proposed binding site I and binding site II, respectively, of the *Os*JAC1-JRL domain. Conserved regions (>50% identity) are shaded in blue. The sequences correspond to proteins from *Zea mays* (*Zm*BGAF), *Triticum aestivum* (*Ta*MCJ1, -2, -3), *Ananas comosus* (*Acm*JRL), *Sorghum bicolor* (*Sb*SL) *Saccharum hybrid* (*Sh*DJ), and *Oryza sativa* (SalT).

**Figure 6 biomolecules-12-01126-f006:**
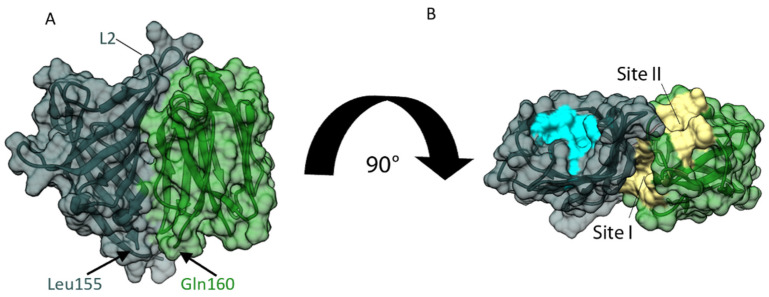
Interaction of individual *Os*JAC1 domains analyzed through protein–protein docking using HADDOCK [34]. Docking was performed with chain A (cf1) of the respective domains. (**A**) Proposed complex of JRL (green) and DIR domain (metallic) shown as ribbon model with additional transparent surface view. A linker with four residues is missing between DIR domain (C-terminus at Leu155) and JRL domain (N-terminus at Gln160). (**B**) Rotated view of panel A. Ligand binding sites of the JRL and DIR domains are highlighted in yellow and blue, respectively. Note that loop L2 of the DIR domain is positioned close to sites I and II of the JRL domain.

**Figure 7 biomolecules-12-01126-f007:**
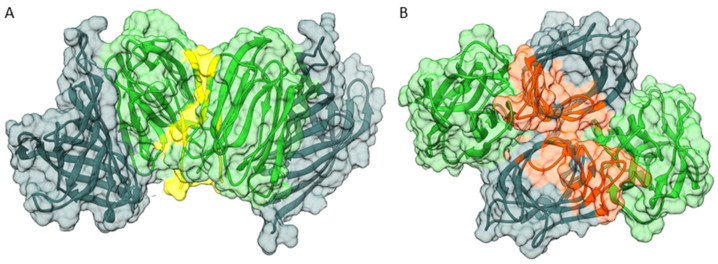
Analysis of *Os*JAC1 oligomerization through protein–protein docking using HADDOCK [34], featuring potential dimers of the full-length protein in transparent surface view with DIR (metallic) and JRL domain (green) ribbon models. (**A**) Active residues were selected according to the interface of the JRL homo-dimer observed in our crystal structures. Interaction is mediated exclusively by the JRL domains (interface highlighted in yellow). (**B**) Active residues were selected according to the interface of the DIR homo-trimer observed in our crystal structures. Interaction is dominated by the DIR domains but also includes significant DIR-JRL contacts (highlighted in orange).

**Figure 8 biomolecules-12-01126-f008:**
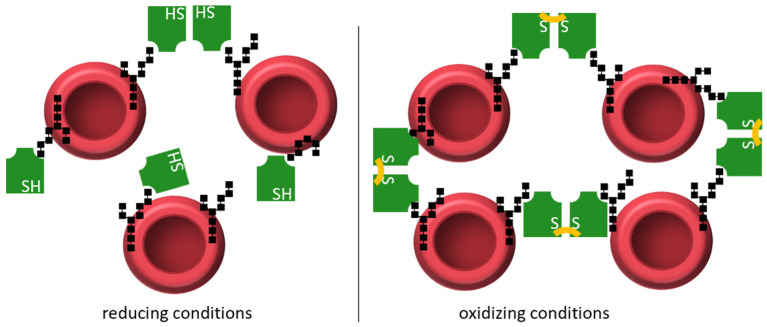
Hemagglutination activity of the *Os*JAC1-JRL domain (green, with respective functional groups of cysteine and cystine indicated as SH and S-S, respectively) interacting with carbohydrate moieties (black shapes) on the surface of erythrocytes. Under reducing conditions (left), the JRL domain forms a sparse interaction network with the erythrocytes, acting as a monomer or a transient dimer. Under oxidizing conditions (right), two JRL protomers are forming an intermolecular disulfide bond (Cys161-Cys161), which leads to a more organized interaction network and increased hemagglutination activity.

**Figure 9 biomolecules-12-01126-f009:**
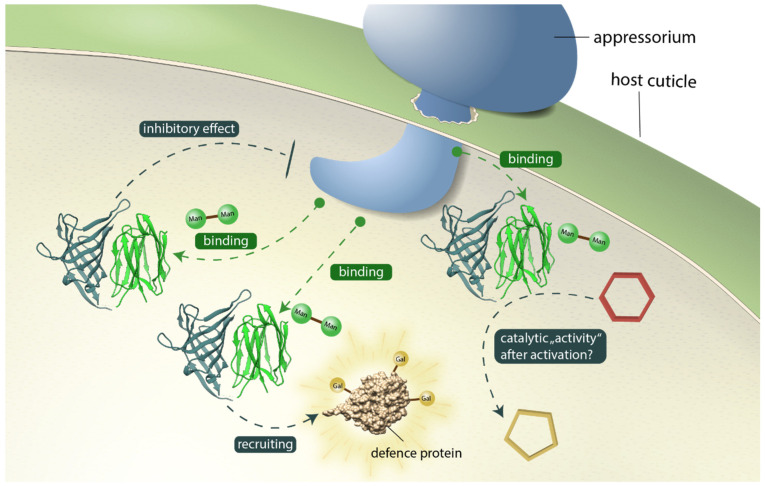
Potential contributions of *Os*JAC1 to pathogen defense at the fungal infection site. The *Os*JAC1-JRL domain (green ribbon) recognizes mannose glycoconjugates related to the membrane or proteins from plant or fungus. Antimicrobial mechanisms of the *Os*JAC1-DIR domain (metallic ribbon) include a direct inhibitory effect, recruitment of other defense-related proteins, and auxiliary enzyme activity. The red hexagon and yellow pentagon represents a putative, unknown compound to be converted.

**Table 1 biomolecules-12-01126-t001:** Experimental conditions for crystallization of the two *Os*JAC1 domains.

Specimen	Protein Buffer	Protein Stock Concentration [mg/mL]	Reservoir Solution	Protein-Tag
JRL-cf1	25 mM sodium phosphate, pH 7.0	10	0.05 M MES pH 6.0, 28% (*w*/*v*) PEG 3350	His
JRL-cf2	25 mM sodium phosphate, pH 7.0	11	0.2 M NH_4_I, 20% (*w*/*v*) PEG 3350	Strep
DIR-cf1	15 mM Tris-HCl, pH 7.4, 1 mM DTT	6.25	0.2 M CaCl_2_, 20% (*w*/*v*) PEG 3350	none ^a^
DIR-cf2	15 mM Tris-HCl, pH 7.4, 1 mM DTT	6.25	0.2 M ammonium dihydrogen phosphate; 0.1 M Tris-HCl pH 8.5; 50% (*v*/*v*) MPD	none ^a^

^a^ His-tag cleaved by thrombin.

**Table 2 biomolecules-12-01126-t002:** Data collection and refinement statistics for the *Os*JAC1 structures discussed in this work.

Specimen	*Os*JAC1-DIR (cf1)	*Os*JAC1-DIR (cf2)	*Os*JAC1-DIR (cf2) w/Galactobiose	*Os*JAC1-JRL (cf1)	*Os*JAC1-JRL (cf2)
PDB code	7R5Z	7YWE	7YWF	7YWG	7YWW
Beamline	ID30A-3 (ESRF)	ID30A-3 (ESRF)	ID23-1 (ESRF)	ID23-1 (ESRF)	ID29 (ESRF)
Detector	EIGER 4M	EIGER 4M	EIGER2 16M	PILATUS 6M	PILATUS 6M
Data collection statistics					
Wavelength (Å)	0.968	0.968	0.775	0.979	0.919
Space group	P 2_1_ 2_1_ 2_1_	H 3 (R3:H)	H 3 (R3:H)	P 2_1_ 2_1_ 2_1_	C 2 2 2_1_
Unit cell					
a, b, c (Å)	49.0, 87.4, 93.7	158.9, 158.9, 47.9	158.1, 158.1, 48.7	34.9, 63.5, 105.9	80.3, 99.6, 93.7
α, β, γ (°)	90.0, 90.0, 90.0	90.0, 90.0, 120.0	90.0, 90.0, 120.0	90.0, 90.0, 90.0	90.0, 90.0, 90.0
Resolution (Å)	43.68–1.75 (1.80–1.75)	45.87–2.15 (2.21–2.15)	45.90–2.60 (2.67–2.60)	54.46–1.10 (1.13–1.10)	52.02–1.40 (1.44–1.40)
Wilson B-factor (Å²)	31.9	60.8	86.5	13.3	22.4
No. reflections	41,085	24,425	13,939	87,556	72,055
Multiplicity	13.7 (13.2)	4.2 (4.4)	5.7 (5.8)	2.3 (1.4)	3.74 (3.72)
Completeness (%)	99.4 (97.6)	99.7 (99.9)	99.9 (100.0)	90.9 (56.7)	92.5 (96.6)
I/σI	10.11 (0.69)	7.83 (0.68)	6.02 (0.51)	5.45 (0.85)	8.80 (0.65)
CC_1/2_ (%)	99.8 (34.6)	99.8 (35.3)	99.5 (38.1)	98.8 (61.2)	99.9 (42.9)
R_meas_ (%)	13.9 (439.7)	12.4 (276.8)	20.4 (325.6)	10.4 (73.0)	4.9 (117.1)
Refinement statistics					
No. reflections	41,050	24,347	13,912	87,550	71,955
R_work_ (%)	19.10	21.78	24.24	15.51	16.87
R_free_ (%)	22.90	25.44	27.77	18.80	19.46
RMSD bonds (Å)	0.006	0.002	0.002	0.007	0.013
RMSD angles (°)	0.863	0.460	0.567	1.031	1.190
<B> [Å^2^] (no. atoms)					
Protein	55.4 (3352)	80.3 (3204)	111.4 (3109)	17.3 (2424)	28.1 (2368)
Water	49.6 (120)	68.7 (26)	97.1 (5)	31.8 (441)	40.1 (299)
Other	47.2 (6)	96.4 (28)	118.5 (90)	13.3 (5)	44.1 (39)
Ramachandran plot					
Favored (%)	96.27	96.24	95.07	97.22	96.25
Allowed (%)	3.03	3.05	4.23	2.78	3.75
Outliers (%)	0.70	0.70	0.70	0.00	0.00

cf, crystal form; RMSD, root-mean-square deviation.

**Table 3 biomolecules-12-01126-t003:** Overview of the hemagglutination assay, using rabbit erythrocytes with *Os*JAC1 and its domains. Data represents the lowest determined protein concentrations capable of agglutinating rabbit erythrocytes under oxidizing [0 mM dithiothreitol (DTT)] or reducing (4 mM DTT) assay conditions.

DTT [mM]	Protein	µg/mL
4	*Os*JAC1	0.98
4	DIR	0.98
4	JRL	15.6–7.8
0	*Os*JAC1	0.98
0	DIR	0.98
0	JRL	1.95–3.9

## Data Availability

Not applicable.

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
