# Peer review of "The Crystal Structure of the Defense Conferring Rice Protein *Os*JAC1 Reveals a Carbohydrate Binding Site on the Dirigent-like Domain"

_biomolecules, 2022, doi:10.3390/biom12081126_

Round 1

Reviewer 1 Report

Huwa et al. report crystal structures of the two major domains from the jacalin-related lectin OsJAC1. The protein family is implicated in plant defense mechanisms. Structures of individual domains of each type are known, but how they are coordinated and might work together is unknown. Only the JRL domain had previously been shown to bind a glycan. Here the DIR domain was shown to also bind glycan by solving the cocrystal, and the coordination of the two domains is proposed on the basis of computational docking of the domains to each other. The domain docking is validated by the large amount of buried hydrophobic surface, utilization of a large DIR-DIR crystal-contact interface, display of the glycan binding sites on the same side of the assembled molecule, and accommodation of formation of a dimer of two assembled molecules, including formation of a disulfide bridge. The accuracy of the individual domain structures were also supported by report of two different crystal forms for each domain. Efforts to crystalize the intact protein were not successful. There was little discussion of the how relevant the model disaccharide glycans might be. What sort of pathogen molecules could they be representing? The glycan activity of both domains was supported by agglutination of RBC but was not increased in the assembled molecule. Could this be due to the irrelevant sugar structures? If it were more appropriate, would additive or synergistic binding be observed? Figure 9 illustrates binding of OsJAC1 to disaccharides, but glycoconjugates are thought to be the physiological ligands.  Does the tetramer structure of OsJAC1 support multivalent binding to polysaccharides and/or branched oligosaccharides?

Author Response

The authors are very thankful to the anonymous reviewers for their helpful comments which definitely improved the quality of the manuscript. Thus, the revised version has some additional comments which explain the experimental results and our conclusions in an easier to penetrate manner.

Comments and Suggestions for Authors

Huwa et al. report crystal structures of the two major domains from the jacalin-related lectin OsJAC1. The protein family is implicated in plant defense mechanisms. Structures of individual domains of each type are known, but how they are coordinated and might work together is unknown. Only the JRL domain had previously been shown to bind a glycan. Here the DIR domain was shown to also bind glycan by solving the cocrystal, and the coordination of the two domains is proposed on the basis of computational docking of the domains to each other. The domain docking is validated by the large amount of buried hydrophobic surface, utilization of a large DIR-DIR crystal-contact interface, display of the glycan binding sites on the same side of the assembled molecule, and accommodation of formation of a dimer of two assembled molecules, including formation of a disulfide bridge. The accuracy of the individual domain structures were also supported by report of two different crystal forms for each domain. Efforts to crystalize the intact protein were not successful. There was little discussion of the how relevant the model disaccharide glycans might be. What sort of pathogen molecules could they be representing?

Please have a look into the first paragraph of the discussion. Here, we added some sentences discussing the origin of the glycan structures and the surrogate character of the tested disaccharides.

The glycan activity of both domains was supported by agglutination of RBC but was not increased in the assembled molecule. Could this be due to the irrelevant sugar structures?

Artefacts because of falsely chosen glycans cannot be ruled out. In the context of the hemagglutination assay the sugars have not been tested e.g. as competitive ligands.

If it were more appropriate, would additive or synergistic binding be observed? Figure 9 illustrates binding of OsJAC1 to disaccharides, but glycoconjugates are thought to be the physiological ligands.

We hope that the new paragraph in the discussion emphasizes our hypothesis that the native ligand is not a simple disaccharide.

Does the tetramer structure of OsJAC1 support multivalent binding to polysaccharides and/or branched oligosaccharides?

This might be. This hypothesis cannot be proven on the given data. Not yet.

Thank you very much for your precious time.

Thomas Classen

Reviewer 2 Report

In this manuscript “Crystal structure of the defence conferring rice protein OsJAC1 reveals a carbohydrate binding site on the dirigent-like domain” The authors provide the structures of two domains of OsJAC1 (JRL and DIR), especially the complex structure of DIR with galactobiose, which provide the novelty founding, however, there are some comments need to be revised. 

1.       When CC1/2 has fallen to around 0.2–0.4, or I/σ to around 0.5–1.5, there is little information remaining, so the high-resolution cutoff needs to recalculate and it is also the reason for the higher Rwork and Rfree to their resolution.

2.       What about the oligomeric state of the JRL domain in the solution, is it also a dimer, and what about DIR, is it a trimer in the solution?

3.       In figure 2A, label the interaction residues.

4.       Page 6, lines 183-184 “Nevertheless, the disaccharide could be clearly identified by newly appearing difference density in all three chains of the DIR domain trimer”. It needs to show the simulated annealing omit map of galactobiose in the complex structure.

5.       What is the Wilson B factor of all the crystal data, especially the structures of the DIR and DIR complex?

6.       Page 6, line 187-189, “In this groove, the non-reducing unit of the galactobiose is precisely fitting and is able to interact through hydrogen bonding (mainly with Gln39 and His30, potentially also His79 and 189 Glu111) and van der Waals contacts (involving Pro32, Trp85, and Val110; see Figure 2 C). How did the authors decide on the hydrogen bond and van der Waals? What is the potential hydrogen bonding, it is not scientific writing? In figure 2C, the green dash line usually shows the hydrogen bond, if not, the author needs to measure the distance between the atoms.

7.       Page 6, lines 191-193, how did the author know the structure of the reducing unit. If it is speculated from the solved complex structure, the author needs to give the figure or explanation about the non-reducing unit of the galactobiose changes to the reducing unit.

8.       In figure S2, L8 shows the different positions from OsJAC1-DIR to AtDIR6 and PsPTS1, and also in figure 2A, galactobiose in the hole close to L8, is that possible, that L8 in OsJAC1-DIR helps to bind with OsJAC1-DIR?

9.       In figure 3 and figure 5, needs to show the full name of every DIR protein in the figure legend.

10.   In figure 6, panels A and B should be of the same size.

11.   It does not need to put a supplementary figure legend to the Figure 1 legend. In some paragraphs, the authors use Figure * at the beginning, and in one paragraph, “Supplementary data of Figures 6 and 7: Table S1: Active residues selected for dock-306 ing. Table S2: HADDOCK scoring results.”, “Table 2–source data in Figure S7. Hemagglutination assay plates” should not appear in the main manuscript. Please re-organize.

12.   In the previous report ref 20, 30,31, DIR liked the proteins also show the trimer crystal structures, and in ref 20, it is shown that DIR-liked protein trimer in solution. So the conclusion “It should be noted that the trimer is so far the only known oligomeric state for DIR and DIR-like proteins after crystallization [20,30,31]” is not correct.  And also the conclusion “Since this trimer is incompatible with the domain arrangement proposed for the full-length protein, we chose to investigate potential dimer states which retain as much of the DIR trimer interface as possible” also is not correct. It needs some more experiments to confirm the oligomeric state for FL OsJAC1.

Author Response

The authors are very thankful to the anonymous reviewers for their helpful comments which definitely improved the quality of the manuscript. Thus, the revised version has some additional comments which explain the experimental results and our conclusions in an easier to penetrate manner.

Comments and Suggestions for Authors

In this manuscript “Crystal structure of the defence conferring rice protein OsJAC1 reveals a carbohydrate binding site on the dirigent-like domain” The authors provide the structures of two domains of OsJAC1 (JRL and DIR), especially the complex structure of DIR with galactobiose, which provide the novelty founding, however, there are some comments need to be revised. 

  1. When CC1/2 has fallen to around 0.2–0.4, or I/σ to around 0.5–1.5, there is little information remaining, so the high-resolution cutoff needs to recalculate and it is also the reason for the higher Rwork and Rfree to their resolution.

The reviewer correctly indicates that shells with low CC1/2 and I/sigI do not strongly contribute to overall information content. However, "little information" does not mean "no information". With the introduction of the CC1/2 metric, a cutoff of 0.3 has been widely accepted as a reasonable choice. While we used XDS, the same criterion is applied by default in the data reduction software AIMLESS for estimation of usable resolution. Also note that, with modern maximum likelihood-based refinement algorithms, inclusion of weak data should not significantly deteriorate the model. Choosing the resolution cutoff to achieve the best overall Rwork/Rfree is considered bad practice.

  1. What about the oligomeric state of the JRL domain in the solution, is it also a dimer, and what about DIR, is it a trimer in the solution?

The oligomeric state was previously examined by dynamic light scattering and gel filtration (Huwa et al. 2021). As a result there is more than one oligomer species for each of the single expressed domains. The DIR domain might be also a trimer in solution.

At the end of the discussion we highlighted this aspect.

  1. In figure 2A, label the interaction residues.

Done

  1. Page 6, lines 183-184 “Nevertheless, the disaccharide could be clearly identified by newly appearing difference density in all three chains of the DIR domain trimer”. It needs to show the simulated annealing omit map of galactobiose in the complex structure.

A simulated annealing omit map has been included in the SI (Figure S9 A) to highlight the presence of the ligand.

  1. What is the Wilson B factor of all the crystal data, especially the structures of the DIR and DIR complex?

Done, Table 1 was supplemented with the Wilson B factors.

  1. Page 6, line 187-189, “In this groove, the non-reducing unit of the galactobiose is precisely fitting and is able to interact through hydrogen bonding (mainly with Gln39 and His30, potentially also His79 and 189 Glu111) and van der Waals contacts (involving Pro32, Trp85, and Val110; see Figure 2 C). How did the authors decide on the hydrogen bond and van der Waals? What is the potential hydrogen bonding, it is not scientific writing? In figure 2C, the green dash line usually shows the hydrogen bond, if not, the author needs to measure the distance between the atoms.

As we stated in the figure caption the interactions have been assessed using LigPlot+ (Laskowski, R.A.; Swindells, M.B. LigPlot+: Multiple Ligand–Protein Interaction Diagrams for Drug Discovery. J. Chem. Inf. Model. 2011, 51, 2778-2786)

We used the term potential H-bond, „ because there are not realized in all chains of the trimer”. This explanation has been added in the paragraphs below figure

  1. Page 6, lines 191-193, how did the author know the structure of the reducing unit. If it is speculated from the solved complex structure, the author needs to give the figure or explanation about the non-reducing unit of the galactobiose changes to the reducing unit.

A detailed view of the carbohydrate binding site on the DIR domain (including a 2Fo-Fc electron density map of the ligand) has been included in the SI (Figure S9 B). This should make the steric arrangement clearer.

  1. In figure S2, L8 shows the different positions from OsJAC1-DIR to AtDIR6 and PsPTS1, and also in figure 2A, galactobiose in the hole close to L8, is that possible, that L8 in OsJAC1-DIR helps to bind with OsJAC1-DIR?

We were speculating about it as well, especially because this loop shows a well-defined stretched conformation. Although we are lacking an experimental structure of the full-length protein, the docking towards a two-domain model (shown in figure 6 and 7) supports this hypothesis. This particular loop is juxtaposed to the other domain supporting an extended interface between DIR and JRL.

  1. In figure 3 and figure 5, needs to show the full name of every DIR protein in the figure legend.

Thank you very much, the species have been added to the caption.

  1. In figure 6, panels A and B should be of the same size.

The zoom effect has been deleted, so that there is a sole rotation of the molecule.

  1. It does not need to put a supplementary figure legend to the Figure 1 legend. In some paragraphs, the authors use Figure * at the beginning, and in one paragraph, “Supplementary data of Figures 6 and 7: Table S1: Active residues selected for dock-306 ing. Table S2: HADDOCK scoring results.”, “Table 2–source data in Figure S7. Hemagglutination assay plates” should not appear in the main manuscript. Please re-organize.

Done

  1. In the previous report ref 20, 30,31, DIR liked the proteins also show the trimer crystal structures, and in ref 20, it is shown that DIR-liked protein trimer in solution. So the conclusion “It should be noted that the trimer is so far the only known oligomeric state for DIR and DIR-like proteins after crystallization [20,30,31]” is not correct.  And also the conclusion “Since this trimer is incompatible with the domain arrangement proposed for the full-length protein, we chose to investigate potential dimer states which retain as much of the DIR trimer interface as possible” also is not correct. It needs some more experiments to confirm the oligomeric state for FL OsJAC1.

I am sorry I do not get the position ‘“It should be noted that the trimer is so far the only known oligomeric state for DIR and DIR-like proteins after crystallization [20,30,31]” is not correct.’. The proteins in ref 20, 30 ,and 31 (old enumeration) are trimers like the one presented here. So we are not claiming to have the first trimeric DIR domain structure, if this was the cause for the misunderstanding.

Regarding your second concern, there is a discrepancy of symmetries, C2 for the JRL-domain and C3 for the DIR-domain, which need to be reconciled in a plausible model for the full-length protein. One solution would be the least common multiple, yet it seems more plausible to assume that at least one of the oligomeric states observed in the single-domain crystals is not realised in the full-length protein. Two hypotheses are presented in figure 7.

However, we took this valuable comment to emphasize this hypothetical character and more experiments must be carried out (“Thus the true oligomeric state remains elusive and needs further experiments for clarification. The best way would be an experimental proof of the quarternary structure (e.g. SAXS) of the full length protein.”)

Thank you very much for your precious time,
Thomas Classen

Round 2

Reviewer 2 Report

The authors already revised some of the issues, but it still has some issues, especially with the structure.

Major issue:

The resolution cutoff of the structures OsJAC1-DIR-cf1(7R5Z), OsJAC1-DIR-cf2(7YWE), and OsJAC1-DIR-cf2 (7YWF) are not right. And the authors mentioned that “choosing the resolution cutoff to achieve the best overall Rwork/Rfree is considered bad practice” is also not right. The resolution cutoff is not decided by Rwork/Rfree, it is decided by many factors of Data collection statistics, like Completeness, I/σ, CC1/2, and Rmeas….,In the structures of OsJAC1-DIR-cf1(7R5Z), OsJAC1-DIR-cf2(7YWE), and OsJAC1-DIR-cf2 (7YWF), in high-resolution shell, I/σ and CC1/2 are too low,  Rmeas are too high, so the final refinement result also shows wrong, because the average B factors are much higher than Wilson B factors. The authors mentioned that “with modern maximum likelihood-based refinement algorithms, inclusion of weak data should not significantly deteriorate the model”, this conclusion is not accurate, what is the “weak data”?

Minor issues:

1.    Page 7, lines 223-226. This sentence together with Fig. 2C and Fig.S10 B, is really confusing. The author wrote Fig.S9 B here, but it should be Fig.S10 B. In Fig.2C, His 30 and Gln39 show as the hydrophobic interaction with galactobiose, but not hydrogen bonding, and with Glu111, it is just 2.2 Å, it is not weakly interaction. In addition, the author can put Fig. S10B in the main Fig. 2.

2.    Page 7, lines 223-235, in this paragraph, the authors mentioned the non-reducing and reducing unit of the structure, so how did the author get two different states from one structure?

Author Response

Major issues:

Please find below a point-by-point treatment of the reviewer’s concerns:

The resolution cutoff of the structures OsJAC1-DIR-cf1(7R5Z), OsJAC1-DIR-cf2(7YWE), and OsJAC1-DIR-cf2 (7YWF) are not right.

There is no such thing as a single „right“ cutoff. The chosen cutoff is subject to general concepts and (a combination of) specific criteria applied by the researcher (discussed in detail below).

And the authors mentioned that “choosing the resolution cutoff to achieve the best overall Rwork/Rfree is considered bad practice” is also not right.

This is an unwarranted statement. As measured intensities get less reliable towards outer resolution shells, the R factors per shell will inevitably increase with decreasing dmin in any given refinement. This implies that overall R factors will also have the potential to increase with the inclusion of additional high-resolution data, even if these data help to slightly improve statistics for the lower-resolution ranges. Hence, the overall R cannot be a valid criterion for the choice of resolution cutoff, which was basically our statement. From the phenix.refine FAQ list (on the choice of resolution cutoff): “This is a contentious subject, and not settled, although it is widely agreed that throwing away usable data simply to reduce the R-factors is not an acceptable practice.”

The resolution cutoff is not decided by Rwork/Rfree, it is decided by many factors of Data collection statistics, like Completeness, I/σ, CC1/2, and Rmeas….,

Here, the reviewer is contradicting his own previous statement. We agree, with the exception that Rmeas is considered obsolete for this purpose because it fails to reflect the precision of the merged data (see below).

In the structures of OsJAC1-DIR-cf1(7R5Z), OsJAC1-DIR-cf2(7YWE), and OsJAC1-DIR-cf2 (7YWF), in high-resolution shell, I/σ and CC1/2 are too low, Rmeas are too high, […]

Again, the reviewer needs to properly support his/her claims; “xyz ist too low“ is not a scientific statement.

First of all, in their paper introducing CC½, Karplus & Diederichs have convincingly shown that Rmeas is not suitable for evaluating the usable resolution range (Science 336, 1030ff). For their primary test data, they find that “the proven value of the data out to 1.42 Å resolution contrasts strongly with the Rmeas and <Ī/σĪ> values at that resolution (>4.0 and ∼0.3, respectively) …” In a later review (Curr. Opin. Struct. Biol. 34, 60ff), they affirm that “… no indicators other than CC½ should influence the high-resolution cutoff decisions for data processing.” and later note that “… many analyses are now using the more generous CC½-based cutoffs (with high resolution Rmeas values as high as ∼1000% and <I/σ>mrgd values as low as ∼0.3) …”. This publication contains a very instructive gedankenexperiment illustrating the value of different data precision metrics.

Based on their own model calculations, Evans & Murshudov (Acta Cryst D69, 1204ff) state “… that by the time CC½ has fallen to around 0.2–0.4, or <I/σ> to around 0.5–1.5, there is little information remaining, but that it would be hard to make a definite rule.” And they conclude: “… it seems that changing the resolution cutoff over a considerable range (e.g. from 2.2 to 1.9 Å) makes only a small difference, so the exact cutoff point is not a question to agonize over, but it seems sensible to set a generous limit so as not to exclude data containing real (if weak) information. There is no reason to suppose that cutting back the resolution of the data will improve the model.”

Our choice of resolution cutoff is in line with these recommendations.

[…] so the final refinement result also shows wrong, because the average B factors are much higher than Wilson B factors.

This statement seems to be based on a misconception. There is no reason to believe that inclusion of a limited amount of weak data would lead to a drastic increase of mean B factors after refinement. In general, mean B values exceeding Wilson B is a very common observation, and there are several reasons for this, including fundamental and software-specific ones. First of all, the Wilson plot is thought to be dominated by the more ordered parts of the structure and therefore tends to underestimate the true disorder. Consequently, the larger the fraction of atoms in highly flexible regions of the model (substantial in the case of the DIR domain structures), the larger a deviation of mean B from Wilson B can be expected, because the fraction of atoms “underappreciated” in the Wilson plot increases. Also note that the Wilson B factor relies on the slope of the linear part of the plot and is thus not very well-determined at least in our 2.6 Å data set. Additional contributions to the apparent discrepancy may stem from the scaling models applied by individual refinement programs. In phenix.refine, e.g., the total model structure factor contains an anisotropic scaling matrix, the trace of which is re-located into the individual atomic displacement parameters in the refined model. As demonstrated by the lead developer (http://www.phenix-online.org/pipermail/phenixbb/2010-August/015564.html), this can cause a substantial (yet non-worrisome) increase in model B factors.

The authors mentioned that “with modern maximum likelihood-based refinement algorithms, inclusion of weak data should not significantly deteriorate the model”, this conclusion is not accurate, what is the “weak data”?

The important point here is that the statement is largely independent of the definition of “weak data”. Another statement from the phenix.refine FAQ list: “In general it may be useful to include additional, weaker data in refinement in the final stages, since the weighting performed by maximum-likelihood refinement prevents these from degrading the model, and may improve it in some cases.” As a result, even pure noise in a high-resolution shell should not be detrimental to the model, although it may lead to an increase in R factors. That said, in our study we have chosen resolution cutoffs such that the outer shells should still contribute significant data, by the criteria outlined above.

Minor issues:

  1. Page 7, lines 223-226. This sentence together with Fig. 2C and Fig.S10 B, is really confusing. The author wrote Fig.S9 B here, but it should be Fig.S10 B. In Fig.2C, His 30 and Gln39 show as the hydrophobic interaction with galactobiose, but not hydrogen bonding, and with Glu111, it is just 2.2 Å, it is not weakly interaction. In addition, the author can put Fig. S10B in the main Fig. 2.

It is a good idea: We now swapped the Figure S10 with Figure 2 of the main document. We kept the former Figure 2 within the SI to have the more abstract representation as well for some readers.

  1. Page 7, lines 223-235, in this paragraph, the authors mentioned the non-reducing and reducing unit of the structure, so how did the author get two different states from one structure?

There could be a confusion in the wording. The "reducing" and "non-reducing" units are expressions for functional groups in glycosides. The glycoside present at C1 as an acetal in a glycosidic bond is the 'non-reducing' end, and the other sugar, which is present as a hemiacetal, is called the 'reducing end' because these groups are Fehling- or Tollens-test positive. It is not a different redox state in a structural superposition or the like.